# The Digital Transformation of the Korean Music Industry and the Global Emergence of K-Pop

**Jimmyn Parc** [1] and **Shin Dong Kim** [2,*]

1  Paris School of International Affairs (PSIA), Sciences Po Paris, France and Institute of Communication Research, Seoul National University, Seoul 150599, Korea; jimmynparc@gmail.com
2  Media School, Hallym University, Chuncheon 24252, Korea
*  Correspondence: shinkim.sk@gmail.com

**Abstract:** There are a number of voices who blame digitization for having a number of negative effects on the music industry including a decline in album sales, copyright infringement, unfair royalty payments, and competition with foreign multinationals. Yet, the global emergence of Korean pop music or K-pop suggests a different narrative, particularly given that its growth was largely unexpected among industry experts. Understanding the key to its international breakthrough can thus produce meaningful lessons for the music industries of other countries for their own further take-off. This constitutes the focus for this paper. Digitization has influenced various sectors of the Korean music industry such as business, society, and consumers. It has also transformed the management focus of the industry from analog to digital, from offline to online, from albums to songs, from specialization to integration, from domestic providers to international suppliers, from audio sound to visual images, from possessing to accessing, and from limited integration to synergistic network. This signifies that embracing technology advancement can enhance the competitiveness of cultural industries.

**Keywords:** digitization; digital transformation; K-pop; Korean music industry; Hallyu; Korean wave; cultural industries

---

In recent years, a number of voices in the music industry have blamed online streaming services like Spotify (Stockholm, Sweden) and YouTube (San Brino, CA, USA) for having a negative effect globally on the music business. In fact, several well-known singers such as Adele, Beyoncé, and Taylor Swift have criticized these streaming platforms as posing a threat to their livelihoods [1,2]. While many have been quick to criticize, few have sought to really understand how technological advancement and digitization have changed culture to this extent; a way of thinking in the industry that has led to few effective approaches being adopted in response to this challenge. However, an alternative path that has proven to be successful exists. Korean pop music or K-pop and its global emergence suggest a narrative on how to fully take advantage of these changes in the music industry (Although the term "industry" generally refers to production and related activities, in this paper it is used to have a broader concept that encompasses production and related activities, as well as consumption).

For this reason, there has been noticeable growing interest in K-pop around the world over the past decade [3–6]. Since its emergence, K-pop has been one of the driving forces behind *Hallyu* or the Korean wave which has translated into the rise of Korean (pop) culture. It covers a diverse range of genres from traditional Korean music to hip hop (rap) and electronic dance music (EDM). Among its many artists, K-pop can be characterized by its "idol" culture with boy and girl bands coupled with its iconic synchronized group dance and mixture of Korean and English lyrics. It is also widely known that K-pop has been extensively diffused by the Internet and its platforms. To provide a sense of its scale, two examples can demonstrate clearly its rising global popularity. The first is the singer PSY's viral hit

"Gangnam Style" which came out in 2012 and for several years was the most-watched video of all time on YouTube. The second is the recent success of the K-pop boyband BTS or Bangtan Boys who won the Billboard Award for Top Social Artist for the third year in a row in 2019, beating well-established singers such as Justin Bieber and Selena Gomez [7]. Furthermore, this band recently achieved its first number 1 on the Billboard Hot 100 with the song "Dynamite".

These landmark achievements stand in stark contrast to the situation that the Korean music industry was faced with in the period before the 1990s. Throughout this time, observers pointed out a wide number of factors that hindered its development, such as the dominance of Japanese music styles, Western influences, government censorship, international pressure to open its market, plagiarism, piracy, and negative domestic market conditions (refer to Parc and Kawashima [8] and Parc, Messerlin, and Moon [9]). In particular, Korea was for a long time regarded as existing outside of the "core countries" in the cultural sector. This would then suggest that the emergence of K-pop in the global market needs to be understood better in terms of the key factors behind its international popularity and their implications that can be useful for other music industries (Some consider this as a result of the Korean music industry's continuous efforts since the 1970s to achieve international popularity, particularly in Japan. In this regard, there were a few scattered cases of artists making a breakthrough: Lee Sung-Ae in the late 1970s, Cho Yong-Pil from the late 1970s to the early 1980s, and Gye Eun-Sook in the 1980s and 1990s. However, they are very different from today's popularity in terms of geographical coverage and the level of impact. It should be noted that other countries, such as Japan and a number of European countries, have also undertaken similar efforts toward the US market. Hence, simply mentioning sporadic moments of going abroad is not enough to explain the international popularity of K-pop).

So far, most analyses on K-pop have drawn upon insufficient explanations such as hybridity and cultural proximity, which are elaborated further in the literature review, or have highlighted aspects that have little meaning elsewhere such as cultural uniqueness or superiority (refer to Otmazgin [10]). Moving away from such approaches that often lack true logic, candidate factors should be carefully analyzed by distinguishing changes that have happened ex ante or ex post for the international emergence of K-pop. Among many explanations, internet-related factors such as the influence of social networking services (SNS) hold some weight but are not specific to Korea.

In order to better understand the emergence of K-pop, an analysis on changes in the Korean music industry should be considered. Thus, this paper links digital technology to the music industry and focuses on identifying the key elements that have fostered the digital transformation of the Korean music industry, which has resulted in the global emergence of K-pop. To support such an approach, a comprehensive and systematic analytical tool based upon Michael Porter's diamond model is utilized. We first review critically the literature that has examined the factors behind K-pop's international coverage. Following this, Porter's diamond model and its logic in this situation is applied. We then analyze the reciprocity between technological advancement and the Korean music industry, coupled with its impact on digital transformation of the industry. Next, we discuss how evolution became generalized along with guidelines for cultural policies in an era of digitization. The paper concludes with a summary of implications to be drawn from the analysis and suggests areas for further study.

## 1. Critical Literature Review

While K-pop has been around for some time (in this case, when K-pop began is debatable. Some view it as emerging after the Korean War, thus in the 1950s when American songs had a significant influence on Korean music [9,11]. Others argue that the beginning of K-pop was with Seo Taiji and Boys, a boy band that debuted in 1992 (see Oh and Lee [12] and Howard [13]). However, these different views do not change the analysis and findings of this paper), its international popularity only became more visible in the 2000s. Given these recent developments, an analysis of K-pop's growing international popularity would be stronger if it were focused on more timely variables.

In looking at the existing studies on this topic, they can be categorized into five elements that reflect their approach: (1) cultural proximity, (2) hybridity, (3) state cultural policies, (4) fandom, and (5) SNS.

*Cultural proximity.* Kim and Ryoo [14] and Sung [15] have noted that Korean cultural products are mostly consumed in East and Southeast Asia. In earlier times, it seemed that the export of Korean products enjoyed more popularity in countries that share Confucian values. This has led many scholars then to argue that the spread of Korean popular music and other texts depends upon cultural proximity. However, Lie [16] has pointed out that Korean cultural goods have also enjoyed widespread popularity in Europe, North America, Latin America, and the Middle East. Further insights in this regard can be provided by the number of views on YouTube as a barometer to measure the diversity of its popularity; this platform provides a wide range of music regardless of it being old or new and it is the most popular online streaming site in the world [17].

Jung and Song [18] present the number of views on YouTube by region based on data from 2011, a year before the viral spread of "Gangnam Style". The fact that a large portion of views are concentrated in Asia would seem initially to support the cultural proximity argument (see Column [a] of Table 1). Although the absolute number is meaningful, the analysis can be different if we consider the number of views per person and also per youngster of each country as they are more likely to be the consumers of K-pop; in other words, data for Columns [d] and [e] are more important than those of Column [a] for a fair comparison. Such an approach is used due to great contrasts among population sizes and demographics around the world which on its own does not accurately reflect the real popularity of K-pop. This new approach used here thus reveals a different picture.

**Table 1.** A new interpretation of the number of YouTube views for K-pop (2011).

| Regions | No. of Views [a] (2011, Unit: 1000) [a] | Populations [b] (2011, Unit: 1000) | | No. of Views/Person | |
|---|---|---|---|---|---|
| | | Total [b] | Age Group (15–29 y.o.) [c] | Total [d] = [a]/[b] | Age Group (15–29 y.o.) [e] = [a]/[c] |
| Asia 1 (excl. China and Korea) | 1,507,325 | 2,828,906 | 763,191 | 0.53 | 1.98 |
| Japan | 423,684 | 128,499 | 20,095 | 3.30 | 21.08 |
| Asia 2 ([Asia 1]–Japan) | 1,083,641 | 2,700,407 | 743,096 | 0.40 | 1.46 |
| North America | 289,271 | 346,251 | 72,501 | 0.84 | 3.99 |
| Europe | 173,862 | 737,851 | 144,163 | 0.24 | 1.21 |
| South America | 119,079 | 597,995 | 158,510 | 0.20 | 0.75 |
| Oceania | 30,820 | 37,498 | 8625 | 0.82 | 3.57 |
| Africa | 9631 | 1,066,410 | 295,584 | 0.01 | 0.03 |

Notes: 1. On the number of views, the original source does not clarify if the Caribbean region is included in the category of "South America". Therefore, in contrast to the original format, the population of the Caribbean region is integrated into South America; 2. Data on China are not included in the Asia category due to unavailability of YouTube in the country; 3. The number of views in Korea and the Korean population are excluded. Data sources: a. Jung and Song [18]; b. United Nations [19].

First, compared to those in "Asia 1" and "Asia 2" (China and Korea are excluded; YouTube is not available in China while Korea is where K-pop originates from), people in North America and Oceania are actually the consumers who viewed K-pop videos the most when the number of views per capita is taken into account (see Column [d]). It is noteworthy that these data are from the period before the success of Psy and BTS and that Japan has been the main target for K-pop bands. Second, this becomes even more evident when focusing on the age group of 15–29 who are considered to be the main consumers of K-pop. When Japan is excluded, youths in North America and Oceania consume

more K-pop than the same age group in Asia, again when the number of views per capita is considered (see Column [e]). In order to better understand this perspective, it is also necessary to point out the fact that Korean entertainment companies promoted K-pop more in Asia, notably in Japan, during its early days. Furthermore, Psy and BTS have gained more popularity in North America than elsewhere. All of these explanations suggest that the cultural proximity argument cannot fully explain how K-pop came to enjoy significant popularity at the global level.

Some may argue that the above analysis does not consider the influence of the Korean diaspora. However, this point is irrelevant when taking into account the ratio of diaspora which we have calculated additionally (see Table 2). In this respect, using data for 2011 from the Ministry of Foreign Affairs of the Republic of Korea [20], the ratio of Korean diaspora in each region's population appears to be around 0 percent. Among these regions, North America has the highest diaspora population which is 0.666 percent of the total population, while Oceania, Europe, Asia (excluding China, Japan, and Korea), South America, and Africa all reach close to 0 percent.

**Table 2.** Presence of Korean diaspora (2011).

| Regions | Total Population [a] (2011, Unit: 1000) | Total No. of Korean Diaspora [b] | Ratio of Korean Diaspora |
|---|---|---|---|
| Asia 1 (excl. China and Korea) | 2,828,906 | 1,205,479 | 0.043% |
| Japan | 128,499 | 913,097 | 0.711% |
| Asia 2 ([Asian 1]–Japan) | 2,700,407 | 292,382 | 0.011% |
| North America | 346,251 | 2,307,082 | 0.666% |
| Europe | 737,851 | 656,707 | 0.089% |
| South America | 597,995 | 112,980 | 0.019% |
| Oceania | 37,498 | 161,038 | 0.429% |
| Africa | 1,066,410 | 11,072 | 0.001% |

Notes: 1. The regional distinction is adopted from Table 1; 2. Data for age groups of Korean diaspora are unavailable. Data sources: a. United Nations [19]; b. Ministry of Foreign Affairs of the Republic of Korea [20].

*Hybridity.* The hybridity between Korean and Western cultures is an element often referred to in many existing studies on K-pop's popularity [21–24]. There are various types of hybridity to consider here, but for this paper we will consider music style and lyrics as they are the most commonly referred to in existing studies. In the first case, musical hybridity is interpreted as foreign music genres and styles that have been mixed with traditional Korean pop music [23]. Indeed, much of this influence came during the 1990s when hip hop, reggae, and Euro dance became popular in the Korean music scene. Second, lyrical hybridity is often understood in terms of the lyrical content being a mixture of English and Korean [24,25]. While these two types of hybridity have enriched the K-pop music scene, a broader understanding of this issue is necessary in order to accurately assess its impact on K-pop and its global popularity.

It is important to recognize that many types of music have been popular in other countries before they reached Korea. For example, reggae was already popular in the Philippines, Thailand, and Japan during the 1980s. Hip hop became mainstream throughout the 1980s and 1990s in the United States, Europe, Africa, and Asia. Euro dance became globally popular from the mid-1990s [26]. Therefore, the hybridity of these music styles with traditional local music should not be considered as K-pop-specific but has been part of a global trend.

Hybridity in regard to the lyrics cannot by themselves explain the popularity of K-pop. English words frequently featured in the lyrics of songs from Hong Kong and Japan long before they showed up in Korea [22,27]. This shows that hybridity of lyrics is by no means specifically characteristic

to K-pop. While this may have helped K-pop to reach a competitive level with other prevailing music in the global market, it is still hard to argue that this hybridization *per se* makes K-pop internationally more popular than others that are also hybridized.

*State cultural policy.* Hong [28], Jin [21], and Kwon and Kim [29] have sought to highlight the importance of the Korean government's cultural policy in the rise of K-pop. Here they have noted how performers have benefited from official financial support to hold concerts and festivals. Such government support, though, only began in the late 2000s when K-pop was already showing signs of international popularity [9,30]. Crucially, however, this was directed toward promoting the "national brand" of Korea by exploiting K-pop's popularity rather than intending to foster its growth [12,31,32]. This argument is supported by an official statement from a government official who stated that the "Korean wave including K-pop has been gaining a lot of international attention and it is culturally beneficial to the national interests, so the state provides full support" [33].

*Fandom.* Several studies have identified the zealous fandom of K-pop as a factor in its international popularity [24,25,34]. Fans of K-pop often open websites for various online activities to support their favorite bands. They routinely play newly released songs of their supporting band on various music sites. Some even externalize by posting billboards in public spaces to express support for their favorite bands or organize crowdfunding projects. The importance of K-pop fandom and its activities to support their favorite bands and create a sense of community should not be overlooked, although a focus on fandom perhaps raises the question: how did they come to know about K-pop, how were they able to access the music? Above all, why were they interested in K-pop in the first place? In this respect, why are the fandom activities for K-pop significantly more visible than those for bands from other countries?

*SNS.* In addressing such questions, Jin [21], Jung [35], and Jung and Shim [36] have all emphasized the importance of digitization coupled with technological development and social media. The power of SNS has undeniably helped K-pop to spread more effectively at a low cost. Indeed, digitization seems to offer the most likely explanation for why the Korean music industry has enjoyed progress and prosperity by considering the correlation between K-pop's emergence and the early introduction of the Internet.

Despite its persuasive view, this approach requires further analysis as well given that the Internet is now the basic infrastructure for many countries around the world. As such, it would be very easy for any international entertainment company to utilize the Internet and SNS to promote their work. Furthermore, the availability of the Internet is not limited to K-pop fans, but also to the fandom of other singers and bands from around the world. In other words, if they wish, they can pursue similar supporting activities for their favorite bands and singers. We can see here that this factor has an insufficient explanatory power and requires rather an in-depth analysis of why K-pop and the Korean music industry have more actively engaged in SNS when compared with their counterparts around the world. As such, a more comprehensive approach is needed to understand how business, society, and consumers have reacted to new information and communication technologies in return.

It is here that we wish to emphasize the structure of the music industry. Regardless of digitization, the music industry inevitably encompasses four main segments: production, distribution, performance, and consumption. By considering this structure, it is easy to understand that the role of the Internet and SNS analyzed in previous studies is superficially related to either distribution or consumption. Therefore, this paper will comprehensively cover all of these four segments in depth, which will help to broaden the understanding of K-pop's international popularity.

## 2. Theoretical Background and Methodology

In order to cover the four main segments of the music industry, an effective tool is required that can deal with them systematically and comprehensively. In this regard, Porter's [37] diamond model provides one of the best tools for such analysis. Based on the concept of competitive advantages, Porter states that competitiveness relies on four interrelated components: (1) *factor conditions*, (2) *demand*

*conditions*, (3) *related and supporting industries*, and (4) *firm strategy, structure, and rivalry*. Factor conditions include a country's position in areas of production, such as the presence of skilled labor or other necessary factors to compete in a given industry; demand conditions involve the size and sophistication of home demand for the industry's product or service; related and supporting industries include the presence or absence of domestic suppliers and an internationally competitive related industry; and firm strategy, structure, and rivalry entail the conditions that govern how companies are created, organized, and managed, as well as the nature and intensity of domestic rivalry [37] (p. 71). Based upon this model, he argues that nations are more likely to succeed in industries or industry segments for which these four components are most favorable.

This can further explain how competitive advantages can be created, enhanced, and sustained. In particular, it is crucial to pay attention to the point that competitive advantages can be created in a given environment and disadvantages can even be transformed into advantages [37,38]. In his study, Porter argues that companies or industries achieve competitive advantage through acts of innovation, either in new technologies and new ways of doing things, or in pursuing existing ideas that have never been vigorously pursued [37] (p. 45). As a result, strategies (or industry and business activities) can develop to fit both industry dynamics and changing environments.

This diamond approach has been utilized to analyze cultural industries, including *Hallyu* generally [39,40], Korean dramas and films [41], K-pop [9], and comparison between J-pop and K-pop [8]. Therefore, concerning the structure of the music industry, this tool is extensive enough to cover production, distribution, performance, and consumption which covers comprehensively the industry and is adequate enough to be applied to an analysis of the Korean music industry.

While this paper adopts the diamond approach and the attribute of each component, we have renamed its four components as follows to allow for greater clarity while still retaining the model: *producers* (*factor conditions*); *consumers* (*demand conditions*); *distributors* (*related and supporting industries*); and *business context* (*firm strategy, structure, and rivalry*). The analysis described in this paper begins with *distributors* and ends by dealing with *business context*, as technological advancement brought about changes in distribution first ahead of other segments and later these changes have affected the business context.

As part of the analysis that explains the digital transformation of the Korean music industry, the narrative employed in this paper is not necessarily in chronological order, but is instead more focused on cause and effect; some changes appeared much earlier yet still have an impact. Moreover, the analysis in this paper is based on qualitative assessments on industrial dynamics and are backed up by existing studies. In particular, sources from media outlets are often utilized as they cover the most recent events related to digitization in the Korean music industry that cannot be found in current academic papers.

As yet, there are few studies that have systematically conducted in-depth research on the digitization of the Korean music industry given its recent nature. This has led to analysis on K-pop being conducted without linking the evidence to one another in a logical fashion. The novelty then of this article is the way in which it seeks the causality that can systematically and comprehensively explain the digital transformation of the Korean music industry when facing digitization.

## 3. Digital Transformation of the Korean Music Industry

This section examines the digital transformation of the Korean music industry responding to the changes of business, society, and consumers in the face of digital technological advancement. As the introduction of the Internet and its "routinization" through smart devices have brought about drastic changes [8], the process and responsiveness have materialized in different ways. As such, it would be more effective to cover these two events separately; they are thus categorized as "introduction" (Digitization 1.0) and "routinization" (Digitization 2.0) of the Internet.

*3.1. Digitization 1.0: Introduction of the Internet*

3.1.1. Distributors: From Analog to Digital

As the emergence of new recording technologies made it easier to copy the latest music in Korea, piracy became more prevalent throughout the 1980s and 1990s. The common practice though was not to copy an entire album but rather a pirated album would contain a playlist of the most popular songs. Known as *kilboard* (an amalgamation of the Korean word for street *kil* and the US entertainment brand Billboard), these compilation albums were widely sold by illegal vendors on the streets of Korea. Despite the fact that this was considered as bad practice by the Korean government who periodically sought to crack down on these vendors, *kilboard* albums were very popular as they featured all the best songs of the time and the price was generally three to five times cheaper than buying an album in a shop. In particular, they became popular among the young generation who were the main consumer of music despite having no principal income sources. This arguably made the *kilboard* vendors effective music distributors while promoting the idea that songs are easily accessible cultural products without having to pay much money.

In the early 1990s, PC-based communication network services were dominant before the Internet became widespread in Korea. Personal digital devices were also introduced during this period, such as MiniDisc players and later MP3 players. Online communities regularly featured personal advertisements from those seeking to exchange music with other users, which increased bonds among consumers of a specific band or singer. With loose regulation of intellectual property rights (IPRs) and ubiquitous piracy, tech-savvy youngsters increasingly extracted songs from CDs and exchanged them with their peers.

In witnessing these changes, the Korean music business began to shift from the analog format toward "digital music" albums. The first effort in this case was initiated in 1998 by Cho PD, a little-known Korean rapper, who established his own record label Stardom (Seoul, Korea). He wrote and composed his first online album *In Stardom* which was uploaded and distributed through an online network in 1998. Featuring eight rap songs as MP3 files, this online album became rapidly popular nationwide. Soon thereafter he released an offline version of *In Stardom* in January 1999. Although this work featured eight new songs alongside the original ones from his previous online album, and was classified by censors as offensive material, it achieved nationwide sales of 500,000 albums and was considered to be one of the most successful cases at that time in Korea. Through these processes, the Korean music industry understood the great potential of this new type of distribution which led to more digital albums being released. Today, many Korean bands release their albums simultaneously as both on- and offline platforms. This transition from analog to digital has facilitated the wider and more effective diffusion of K-pop in the era of digitization when compared with export of physical albums.

3.1.2. Producers: From Offline to Online

Led by groundbreaking groups such as Hyun Jin-young and Wawa and Seo Taiji and Boys, the Korean music industry underwent a significant change in terms of music style moving from ballads to rap music in the early 1990s (Hyun Jin-young was trained under Lee Soo-man who was the founder of SM Entertainment). In other words, the direction of the musical influence was unilateral, from international to domestic. Witnessing the shift from offline to online markets, Korean businesses anticipated not only an increase in domestic market size, but also an expansion of market scope. In other words, a bilateral shift. This trend became a reality following the sudden success of the Korean dance music duo Clon who enjoyed widespread popularity in Taiwan during the late 1990s. Thus, the music style transformations during this period accelerated further to meet the changing domestic and international tastes.

Under these circumstances, entertainment companies began to hire bilingual musicians to sing and rap in English, which helped to penetrate foreign markets as well as to appeal to international fans. In this context, the largest entertainment company in Korea, SM Entertainment (Seoul, Korea,

hereafter SME), formed the girl group S.E.S. in November 1997 whose members could speak English and Japanese. Initially, Korean companies focused on the Japanese market but soon expanded to recruit Korean American singers in the late 1990s, and other performers from China and Thailand as the years passed. This strategy has also been adopted by late-comers such as YG Entertainment (Seoul, Korea, hereafter YGE) and JYP Entertainment (Seoul, Korea, hereafter JYPE).

As the market has functions bilaterally between domestic and international through on- and offline platforms, it is not just the consumption market that has expanded, but also the way music is produced. In this regard, Korean companies and their bands and singers collaborate with international musicians in order to enlarge their fandom internationally as well as to widen their music genres. For example, American musicians such as Flo Rida and Diplo collaborated with Korean musicians such as BoA and G-Dragon in the late 2000s. This trend continued to be strengthened throughout the 2010s as the Nordic pop group Bracelet collaborated with the Korean boy band B.A.P while John Legend released a duet with Wendy from Red Velvet. In particular, BTS has been one of the leaders in such collaboration. This group has teamed up with a range of well-known American musicians, such as The Chainsmokers, Desiigner, Halsey, Lauv, Nicki Minaj, Steve Aoki, and Warren G. This endeavor has helped K-pop to further diversify its genres and to introduce itself to the local fandom of well-established collaborators. Hence, these efforts have contributed toward expanding K-pop's influence and market through both on- and offline platforms. All in all, this has helped to increase its international popularity.

### 3.1.3. Consumer: From Album to Song

The rapid digitization of the music industry changed consumer purchasing behavior across the world. Instead of buying a CD album, consumers in the 2000s began to purchase (or download) only the popular songs from online music platforms or stores [42]. In other words, instead of buying an album that has only one or two appealing songs while the others are less so, consumers collect only those songs that they enjoy more. In fact, this selective behavior appeared in Korea much earlier than other countries as Korean youngsters extracted songs in MP3 format from CDs and traded them online. For consumers, this process divorced songs from their albums and changed the way in which they engaged with music. Rather than simply criticize this behavior, Korean music producers instead began to reconsider their own production strategies in response. The approach they adopted in this sense was to increasingly focus their efforts on producing just a few standout tracks rather than utilizing all their resources to create multiple tracks for an album.

Because they were now concentrated on producing just a few tracks, such a limited number of songs had to be of the highest quality. To achieve an internationally competitive standard, Korea's entertainment companies began to forge new contacts with leading foreign composers from the late 1990s. For example, several songs performed by S.E.S. were in fact composed by Japanese and Finnish song writers or were even an official cover version. For example, their widely popular song "Dreams Come True" (1998) was actually a cover of "Like a Fool" by the Finnish girl group Nylon Beat. With this successful formula, these entertainment companies went on to establish a tight network with foreign song writers, notably those from Scandinavia. SM leads the way in such efforts and has a collaborative network with 864 international songwriters as of June 2020 [43]. This endeavor helped them to enhance their musical quality and has allowed K-pop to be competitive in the global market. Part of this effort has been externalized as international collaboration which explained the precedent section.

Another interesting point in this regard can be found with the Japanese music industry. Since the mid-1980s, its artists have been releasing what is known as "single albums", a format which contains only one or two songs as part of efforts to increase sales. This single-album strategy was adopted in Korea. Later when combined with the Internet, it resulted in Korean companies producing the more profitable "digital single albums"; the first of this kind being released by the Korean singer Seven in 2004. This strategy has been further advanced with mini albums and compilation albums; a retro-trend that happened in the past but armed with digitization and the Internet. One notable example is BTS.

This band has released a number of single, mini, regular, repackaged (or compilation), and special albums by putting several title songs at the heart of these albums (The classification for a "single album" has not been clearly defined. However, in general, a single album has 1–4 main songs, a mini album has around 5 songs, and regular albums have approximately 10 songs. Repackaged albums add a few new songs to the original album, whereas special albums refer to albums of limited editions and ones that are made for a special purpose, often including gifts such as concert tickets). In contrast with past practices, fans today purchase albums in order to support their favorite singers and groups rather than to enjoy them as a musical experience [44]. All of these efforts contribute to a larger consumption of more appealing songs than before.

### 3.1.4. Business Context: From Specialization to Integration

The distribution of albums is closely related to their sales, which is a conventional source of revenue. When distribution does not effectively function due to negative factors like when piracy was widespread in Korea, other parties will seek to overcome this malfunctioning segment through alternative means as seen with the production side. Under these circumstances, the role of music sources such as radio and TV has become that of new distribution outlets as they help to increase the commercial reputation of singers and bands.

In particular, as the music market size increased significantly, the efficiency in producing in quantity and distributing music effectively has become an even more crucial factor. Several pioneers such as Lee Soo-man recognized that there would be a promising future for entertainment companies in Korea and therefore began to train up many aspirants with great potential to specific requirements instead of going through the lengthy and somewhat unpredictable process of talent scouting (It is also known that Lee Soo-man benchmarked the success of Johnny & Associates (Tokyo, Japan), one of the largest Japanese entertainment conglomerates in the Japanese music industry). In other words, these entertainment companies integrated various functions from production and distribution to training and management. Given that these online networks have functioned as effective distribution channels and outlets, they have not only anchored the position of entertainment companies in the Korean music industry but have also helped K-pop to spread abroad.

As domestic and international fans are able to easily enjoy music and music videos online, they are then keen to embrace stimulating experiences such as on-site performances. Since Korean idol groups have focused on both music and dance, this fact has made their concerts more of a spectacle [45,46]. It is important to note that such live shows cannot be so easily copied or found online and they have been identified by music companies as a key source for revenues in the face of declining offline music sales. In seizing this opportunity, companies are increasingly looking to incorporate a section that deals with on-site performance. A good example in this case is the merger of SME with the event organizer Dream Maker (Seoul, Korea) in 2005 which reflects the substantial profitability of holding concerts. Such vertical integration has become a more prevalent characteristic in the Korean music industry and has contributed to vibrant dynamics.

### 3.2. Digitization 2.0: Routinization of the Internet

### 3.2.1. Distributors: From Domestic Providers to International Suppliers

As piracy was prevalent in Korea, local streaming service providers had to be careful about increasing the subscription fees and the price of songs that they were offering. If they were to charge too much then the younger generation of consumers who are the majority would simply switch back to illegal downloads [47,48]. In contrast, if their appropriate level of contracted royalty does not meet the expectation of the musicians and their managing companies, these service providers would have difficulties to secure a pool of K-pop songs. This complex situation places the Korean service providers in a delicate situation. Furthermore, as the routinization of the Internet became greater and the power of SNS and YouTube increased, Korean streaming service providers were concerned about

the entry of foreign service providers such as Amazon (Seattle, WA, USA), iTunes (Cupertino, CA, USA), and Spotify. However, Korean entertainment companies had a different view toward these players as they had already experienced the effective and wide distribution networks of international platforms such as Facebook (Menlo Park, CA, USA) and YouTube.

Although these international streaming service providers have shown keen interest in the Korean market, they face several "invisible barriers," which led them to initially hold back their entry. First, the younger generation of consumers prefer to listen to K-pop rather than foreign songs; for this, Korean service providers have comparative advantages [47,48]. Second, subscription fees in Korea are much lower than other countries. The result is that several local providers such as Melon (Seoul, Korea) and Genie (Seoul, Korea) have enjoyed a cost advantage which has forced international service providers to carefully consider how they will enter the Korean market.

Despite this unique environment with the aforementioned disadvantages, foreign service providers have more incentive to secure a pool of K-pop songs due to its growing global popularity and demand around the world. This makes them willing to purchase K-pop at a higher price which creates more revenue for the musicians and entertainment companies than what local platforms and service providers pay them [47]. Many of these Korean entertainment companies recognized this as an opportunity to distribute K-pop abroad more easily and effectively because they are confident about the dissemination power of these international service providers.

### 3.2.2. Producers: From Audio Sound to Visual Images

Visual images have long played an important role in pop music as technological advancement has progressed. There are many examples of famous television appearances from Elvis Presley's dance moves to Michael Jackson's glittering costumes; the first case was broadcast in black-and-white television whereas the latter was in color. The appearance of dedicated music channels such as MTV (New York, NY, USA) pushed further the integration of audio sounds and visual images as music videos sought to be more appealing by featuring a storyline within their contents. The screens of smart devices have become another supplementary outlet for music videos as the Internet became routinized in the late 2000s.

In this environment where visual images play a significant role, Korean entertainment companies have focused on enhancing the choreography of their bands such as group dance. This has helped them to satisfy the visual appetite of consumers [8]. Obviously, many do not understand this aspect, but often rashly criticize the thick make-up among singers and disdain the well-organized group dances of these idol groups as "manufactured" art, while they are often considered to be the symbol of K-pop itself. In sum, with advances in audio/visual technology, the trend in the music industry has changed from "music to listen to" to "music to listen to and watch," a process which Korean entertainment companies have actively adapted to.

Recently, the integration of audio sounds and video images has been further developed in a more sophisticated way. In a similar way to the Marvel Cinematic Universe, often known as MCU, Big Hit Entertainment launched the BTS Universe or BU in 2015 to promote its band. In other words, a series of storylines continues not only in a single song, but also across several songs and albums. Consequently, BTS has inserted many subplots and Easter eggs into their songs and music videos as a subtle way to draw in the audience more. Among the many examples, this can be seen with the songs "Save Me" (2016) and "Fake Love" (2018) which were released two years apart from each other but feature a connection. The title of "Save Me" appears written on a wall in the "Fake Love" music video at the 4:51 time mark. If this image is reversed as shown in Figure 1, it spells out "I'm Fine". This created much speculation among BTS fans. Some guessed that it was a sign telling its fans that the group is fine despite online attacks from BTS haters or some played out difficulties in the BU storyline, while others took it as a hint of a new song which will be released in the near future. As predicted, the song "I'm Fine" was released in August 2018 and it is treated as an extension of "Save Me" as it begins with the seven members of the boyband standing in the same position as when "Save Me" ends.

These endeavors have made K-pop more interesting and appealing than their counterpart around the world.

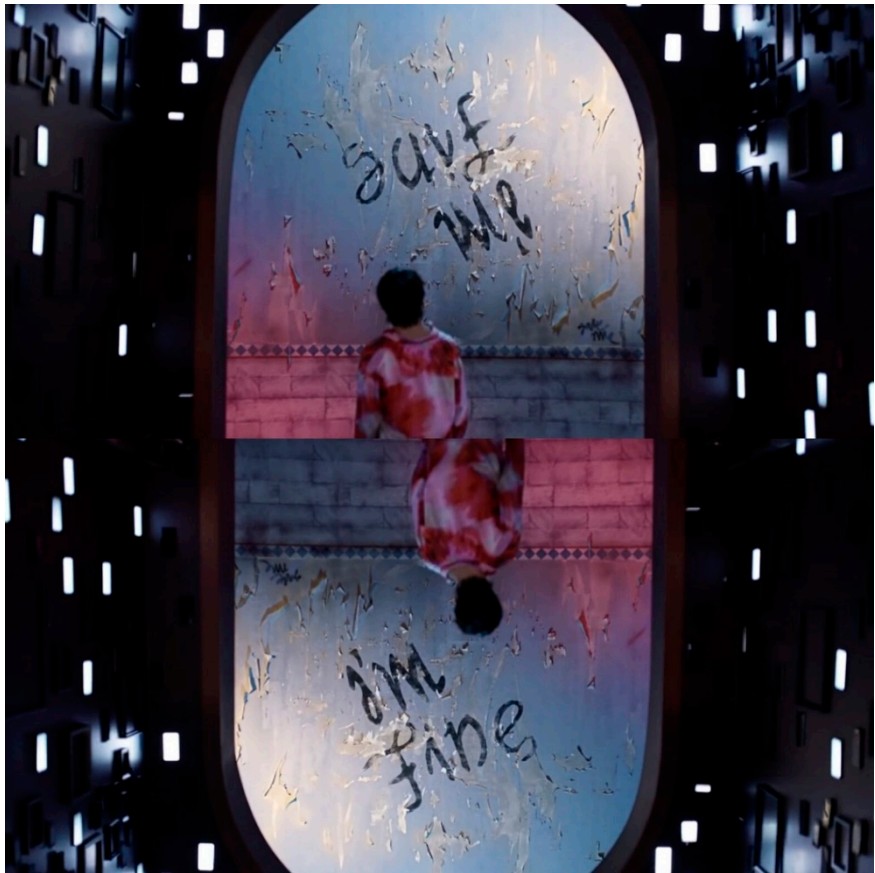

**Figure 1.** The images of "Save Me" and "I'm fine". Image source: BTS's "Fake Love" music video (4:51), [49].

### 3.2.3. Consumers: From Possess to Access

As technological advancement changed the music industry from analog to digital, consumers collected music files instead of physical albums. Given the limitations brought on by computer hard drives, consumers began to increasingly turn to virtual storage services such as Webhard (Seoul, Korea) and Soribada (Seoul, Korea) in the early 2000s. This allowed users to download and exchange music as well as to save it in their virtual folders.

Although these services were popular among younger generations, the emergence of smart devices shifted the focus among consumers further. During this time, they underwent a dramatic transformation from "possessing music" on virtual storage platforms or MP3 players to "accessing music" via software or music streaming through smart devices [11]. Although IFPI [50] and IPSOS and IFPI [42] have all described this as a recent global trend, streaming services actually appeared earlier in Korea than other countries. For example, the online music streaming service Melon was launched in 2004, several years before Spotify appeared on the scene [51].

Facing these changes, Korean entertainment companies understood from the beginning the importance of SNS and internet platforms as effective tools for disseminating music under "loose" copyright practice in Korea. For example, they were among the first to establish official YouTube channels in order to promote newly released songs more effectively. Hence, a large number of (new and old) K-pop songs have become available online. As online diffusion through these internet platforms is wider and faster than sales of physical albums, international fans have been able to easily access K-pop through various sources on the Internet [9]. Furthermore, this has become a way in which Korean

entertainment companies have learned about the preferences of fans around the world, thus it leads them to produce more internationally appealing music. As a result, the consumption and size of K-pop have significantly increased domestically and internationally.

### 3.2.4. Business Context: From Limited Interaction to Synergistic Networks

Korean companies have recognized the way in which the function of music has transformed from end products to promotional products in the era of digitization, this can be a starting point for other (un)related business, such as fashion, restaurants, movies, and advertising as well as on-site performances. This integrated diversification is also a way to establish stronger business portfolios that can help secure more stable revenue sources while spreading the possible risks that it may cause. For this, Korean entertainment companies established subsidiaries that specialized in specific sectors. For instance, SME has Esteem (Seoul, Korea, for fashion), SM Culture & Contents Co., Ltd. (Seoul, Korea, for dramas and variety shows), SM Food and Beverage Co., Ltd. (Seoul, Korea), and SM Town Travel (Seoul, Korea). In the same vein, YGE has Moonshot (Seoul, Korea, for cosmetics), YG Sports (Seoul, Korea, as golf agency), and YG STUDIOPLEX (Seoul, Korea, for drama production). Finally, for their part, JYPE has JYP Foods Inc. (New York, NY, USA), JYP Pictures Co., Ltd. (Seoul, Korea), and JYP Publishing Co., Ltd. (Seoul, Korea).

Such a diversification in business operations has pushed these entertainment companies to target stars who have differentiated talents that fit these varying sectors more efficiently. To meet the new market demands, these companies have placed more effort and investment into fostering promising groups and singers by utilizing revenues gained from previously successful acts within the same company [5,52]. For example, with the profits earned by successful singers and groups such as BoA and Girls' Generation, SME was able to invest in future talent. In the same way, entertainment companies have invested in diversified business by using revenues from various cash cow sectors. This is similar to the *chaebols* or conglomerates that emerged in Korea as a way to interact with both related and seemingly unrelated sectors, known as "chaebolization". As time goes by, these businesses have shown a patchy performance and have had to be restructured in order to possess more efficient and synergetic networks.

These Korean entertainment companies have also sought to expand more aggressively abroad in order to broaden their geographical scope of business activities. Thus, they have established overseas joint ventures and/or subsidiaries in countries such as China, Japan, and the United States. In this way, BoA from SME and Wonder Girls from JYPE branched out into the US music industry and achieved a degree of recognition. By contrast, PSY and his agency YGE worked with Scooter Braun and his Schoolboy Records (Santa Monica, CA, USA) in order to make the song "Gangnam Style" become a viral hit. Since then, Korean entertainment companies have collaborated more with local companies or agencies when needed, instead of establishing or working with local subsidiaries. In going through these synergetic networks, Korean entertainment companies have become more internationally competitive.

## 4. Discussion

This paper has shown how digitization has brought about an impact upon almost all segments of the music industry. Such a transformation has also changed the conventional process of music production, distribution, performance, and even consumption. Given these new dynamics, the prosperity of a country's music industry depends on how each segment in the industry chain responds to such transformations while enhancing more effectively the core functions of each segment; good quality of music in large quantity, efficient and wider distribution, and mass but sophisticated consumption. In this regard, the Korean music industry can be an interesting case example to explore.

There were two phases in terms of digital transformation. The first one is with the introduction of the Internet and the latter is with its routinization. Based on the previous analysis, the first phase is captioned as "Series A", whereas the latter is labeled as "Series B" in Table 3. The Korean music

industry has responded more actively to digitization by embracing these changes and has now taken advantage of it. Looking at Table 3, it provides a summary of the whole digital transformation of the Korean music industry systematically and comprehensively. Although a sequence of events can be found in Table 3, it is noteworthy that all these developments have occurred pretty well much simultaneously and within a relatively short period of time.

**Table 3.** Digital transformation in the Korean music industry.

|  |  | **Changes** | **Responsiveness of the Korean Music Industry** |
|---|---|---|---|
| Distributors | Hardware (A-1) | Analog ⇒ Digital | • Distribution of MP3 files via the Internet (early 2000s)<br>• Digital album (1998) |
| Distributors | Software (B-1) | Domestic providers ⇒ International suppliers | • International streaming service providers (2010s)<br>• Easier access to the global market (2015) |
| Producers | Basic (A-2) | Offline ⇒ Online | • Globalization of idol groups (late 2000s)<br>• International collaboration (late 2000s) |
| Producers | Advanced (B-2) | Audio sound ⇒ Visual images | • Make-up and choreography; group dance (early 2000s)<br>• Synergetic use of audio sounds and visual images (2010s) |
| Consumers | Size (A-3) | Album (bundle of songs) ⇒ Song (a piece of music) | • Enhanced musicality (early 2000)<br>• Digital single album and its extension (2010s) |
| Consumers | Quality (B-3) | Possessing ⇒ Accessing | • Virtual storage, e.g., p2p (2000)<br>• Internet platforms (early 2000s) |
| Business context | Structure (A-4) | Specialization ⇒ Integration | • Establishment of entertainment companies (late 1990s)<br>• Verticalization and M&As (2000s) |
| Business context | Synergy (B-4) | Limited interaction ⇒ Synergetic network | • Music as promotional tool, on-site performance (2000s)<br>• "Chaebolization" (early 2010s) |

There are two key ways in which Table 3 provides a useful overview: (1) it summarizes critical factors systematically and comprehensively in a way that provides a better understanding of the transformation of the music industry when confronting digitization; (2) practitioners and policy makers can use this table to see where hindrances exist and how to overcome them by analyzing the case of the Korean music industry.

Some advocates for creative industries argue that culture should be protected and conserved, a point which ignores the fact that it can further evolve through changes like technological advancement. Moreover, they often emphasize the importance of state aid for the industry as it is closely linked to national culture and identity. While this paper does not deny the need for this kind of support, the case of the Korean music industry highlights different lessons, and this should be taken into account when policy makers establish cultural policies. To sum up, the Korean music industry has evolved by overcoming its disadvantages and enhancing existing advantages while understanding and embracing comprehensively the changes brought on by digitization and technological advancement. This all makes their entrepreneurship more dynamic and business activities more innovative which can produce an outstanding performance.

## 5. Conclusions

When the Korean music industry began to boom in the early 1990s, many observers expressed concerns about the future of the industry as it faced an environment that had been "devastated" by digitization. At the time, the industry was confronted by a multitude of challenges including the loose practice of IPRs, rampant piracy through the Internet, and the lifting of the ban on Japanese cultural products. Some even argued that there would be no future for the Korean music industry. However, despite the fact that these issues have not been entirely solved, the Korean music industry was able to survive and flourish further in the global market. In seeking to explain this success against the odds,

many scholars have pointed out Korea-specific factors which are not available in other countries nor can be found easily in other countries while being new in Korea.

Such arguments may explain how K-pop was able to reach a similar level with the pop music from other countries, but still cannot fully explain why it has become more distinguishable and popular than its counterparts. In this regard, a rigorous analysis on the evolution of the Korean music industry has very important lessons. It will help to identify the factors that can be applied in countries that wish to develop their music industry.

Instead of sticking to traditional structures and practices, the Korean music industry understood the transformation that came with digitization such as from analog to digital, from offline to online, from albums to songs, from specialization to integration, from domestic providers to international suppliers, from audio sound to visual images, from possessing to accessing, and from limited interaction to synergetic network. More importantly, the Korean music industry has embraced these changes by adjusting their activities. This adaptation process has significantly enhanced the viability of K-pop in the global market during this era of digitization.

There are two meaningful implications from the analysis of the Korean music industry presented in this paper. First, the transformation due to digitization should be comprehensively understood instead of merely focusing on a few sectors. By understanding these aspects more systematically, meaningful implications can be drawn for other countries. Second, in understanding these transformations, the Korean music industry did not wrestle with digitization but rather embraced it. As a result, K-pop fans all over the world were able to easily access the music produced, and Korean entertainment companies and their idol groups would then try to meet the needs of their international fan base.

Since the analysis in this paper focuses on comprehensiveness to understand the increasing international popularity of K-pop, the transformation of business, society, and consumers in the Korean music industry when faced with digitization is clearly introduced and systematically scrutinized. Therefore, each element would be of interest for further studies, especially given the fact that digitization and technological advancement can be found all over the world. Furthermore, it can be interesting to assess the current competitiveness level of the Korean music industry after it has embraced digitization vis-à-vis its international counterparts. Addressing these topics can help enhance the rigor of the related studies in order to help society achieve true cultural diversity and competitiveness.

**Author Contributions:** J.P. designed this research and contributed to writing/revising the main parts of this article. S.D.K. commented on and revised this article. All authors have read and agreed to the published version of the manuscript.

**Funding:** Jimmyn Parc's work was supported by the Laboratory Program for Korean Studies through the Ministry of Education of the Republic of Korea and the Korean Studies Promotion Service of the Academy of Korean Studies (AKS-2015-LAB-2250003); Shin Dong Kim's work and the article processing charges (APC) for this paper were supported by the National Research Foundation of Korea (NRF-2015S1A5B4A01037022).

**Conflicts of Interest:** The authors declare no conflict of interest.

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
