# Peer review of "The Digital Transformation of the Korean Music Industry and the Global Emergence of K-Pop"

_sustainability, doi:10.3390/su12187790_

Round 1
Reviewer 1 Report
The article discuss functioning of K-pop industry in digital environment. It demonstrates scientific approach and relevance, but it should be improved according to the comments in text.

Author Response
We appreciate the constructive comments and encouragement from reviewers. They were very useful to help us improve the quality of this paper. We have incorporated most of the suggestions and hope that our responses and revisions will be satisfactory. What follows presents our responses and the changes we have made.
Comments and responses
Reviewer 1: The article discuss functioning of K-pop industry in digital environment. It demonstrates scientific approach and relevance, but it should be improved according to the comments in text.
- Thank you for your comments. They were very helpful. We checked your detailed remarks in the file and incorporated most of them.
Comment 1: Brief description of K-pop music, lyrics, dance and visual is needed here. (Line 34, Page 1)
- Thank you, we have modified the text as follows:
Before:
For this reason, there has been noticeable growing interest in K-pop around the world over the past decade (Han, 2017; Hjorth, 2008; Oh and Park, 2012; Otmazgin and Lyan, 2014). To provide a sense of its scale, two examples can demonstrate clearly its rising global popularity. The first is the artist PSY’s viral hit “Gangnam Style” which came out in 2012 and for several years was the most-watched video of all time on YouTube. The second is the recent success of the K-pop boyband BTS or Bangtan Boys who won the Billboard Award for Top Social Artist for the third year in a row in 2019, beating out well-established singers such as Justin Bieber and Selena Gomez (Billboard, 2020).
After:
For this reason, there has been noticeable growing interest in K-pop around the world over the past decade (Han, 2017; Hjorth, 2008; Oh and Park, 2012; Otmazgin and Lyan, 2014). Since its emergence, K-pop has been one of the driving forces behind Hallyu or the Korean wave which has translated into the rise of Korean (pop) culture. It covers a diverse range of genres from traditional Korean music to hip hop (rap) and electronic dance music (EDM). Among its many artists, K-pop can be characterized by its “idol” culture with boy and girl bands coupled with its iconic synchronized group dance and mixture of Korean and English lyrics. It is also widely known that K-pop has been extensively diffused by the Internet and its platforms. To provide a sense of its scale, two examples can demonstrate clearly its rising global popularity. The first is the singer PSY’s viral hit “Gangnam Style” which came out in 2012 and for several years was the most-watched video of all time on YouTube. The second is the recent success of the K-pop boyband BTS or Bangtan Boys who won the Billboard Award for Top Social Artist for the third year in a row in 2019, beating out well-established singers such as Justin Bieber and Selena Gomez (Billboard, 2020). Furthermore, this band recently achieved its first number 1 on the Billboard Hot 100 with the song “Dynamite.”
Comment 2: references
1/ Line 42 on page 2
- It has been properly added.
2/ Line 50 on page 2
- It has been properly added.
3/ Line 77 on page 3
- There are too many scholars within this topic and in the context of the previous version, we do not feel it is necessary to add new references.
4/ Line 92 on page 3
- This is not based on existing studies, but explains a finding based on Table 1 in this paper.
Comment 3: corrections of English (Line 68, Page 3)
1/ Line 69 on page 2
- It has been properly corrected.
2/ Line 322 on page 8
- It has been properly corrected.
Comment 4: other platforms, CD sale, broadcasting? (Line 80, Page 3)
- It has been modified as follows:
Before:
Cultural proximity. Kim and Ryoo (2007) and Sung (2010) have noted that Korean cultural products are mostly consumed in East and Southeast Asia. In earlier times, it seemed that the export of Korean products enjoyed more popularity in countries that share Confucian values. This has led many scholars then to argue that the spread of Korean popular music and other texts depends upon cultural proximity. However, Lie (2012) has pointed out that Korean cultural goods have also enjoyed widespread popularity in Europe, North America, Latin American, and the Middle East. Further insights in this regard can be provided by the number of views on YouTube as a barometer to measure the diversity of its popularity.
After:
Cultural proximity. Kim and Ryoo (2007) and Sung (2010) have noted that Korean cultural products are mostly consumed in East and Southeast Asia. In earlier times, it seemed that the export of Korean products enjoyed more popularity in countries that share Confucian values. This has led many scholars then to argue that the spread of Korean popular music and other texts depends upon cultural proximity. However, Lie (2012) has pointed out that Korean cultural goods have also enjoyed widespread popularity in Europe, North America, Latin America, and the Middle East. Further insights in this regard can be provided by the number of views on YouTube as a barometer to measure the diversity of its popularity; this platform provides a wide range of music regardless of them being old or new and it is the most popular online streaming site in the world (Jung, 2017).
Comment 5: Korean diasporas (Page 3)
1/ What ethnic populations in North America and Oceania? In which percent is diaspora present?
- First, this paper is about the transformation of the Korean music industry. In this regard, the exact demographics in North America and Oceania are not relevant for this section. There are a significant number of media articles, academic articles, and YouTube video clips that clearly demonstrate how regardless of their ethnic background, fans enjoy K-pop.
- Second, Table 1 is based on the data of Jung and Song (2010). In their data, the number of Korea diaspora is not considered.
- Third, in the early period, there are a few who argued that K-pop was only consumed by Korean diaspora and/ or ethnic minorities. Again, there are a large number of new media articles, academic articles, and YouTube video clips that clearly demonstrate that this is irrelevant.
- Regarding the number of Korean diaspora, the population ratio in all the regions is close to 0 percent as shown in the table below (data for age groups are not available):
|
Regions |
Total population (2011, Unit: 1,000) |
Total no. of Korean diasporas |
Ratio of Korean diasporas among local population |
|
Asia (excl. CN and KR) |
2,828,906 |
1,205,479 |
0.043% |
|
Asia excl. CN, JP, KR |
2,700,407 |
292,382 |
0.011% |
|
Japan |
128,499 |
913,097 |
0.711% |
|
North America |
346,251 |
2,307,082 |
0.666% |
|
Europe |
737,851 |
656,707 |
0.089% |
|
South America |
597,995 |
112,980 |
0.019% |
|
Oceania |
37,498 |
161,038 |
0.429% |
|
Africa |
1,066,410 |
11,072 |
0.001% |
- If we consider the presence of Korean diaspora, as shown above, the ratio is insignificant. Additionally, if the diaspora is an important factor for the popularly of K-pop, then a general trend should be observed, unfortunately we cannot identify any noticeable result. For example, although Europe has a larger percentage of Korean diaspora than South America, the number of views per person which is shown in Column [4] is very similar.
- More importantly, it is necessary to examine if the number of Korean diaspora has changed significantly before and after the emergence of K-pop or if this variable has been consistent. This fact clearly shows that the influence of the Korean diaspora (in terms of number) is not much related to the global emergence of K-pop.
2/ Cultural proximity actually explains this remote popularity of K-pop in case of diaspora communities!
Table 1. A new interpretation on the number of YouTube views for K-pop (2011).
|
Regions |
No. of views (2011, Unit: 1,000)a [1] |
Populations (2011, Unit: 1,000)b |
No. of views per person |
||
|
Total [2] |
Age group (15-29 y.o.) [3] |
Total [4]=[1]/[2] |
Age group (15-29 y.o.) [5]=[1]/[3] |
||
|
Asia (excl. CN and KR) |
1,507,325 |
2,828,906 |
763 191 |
0.53 |
1.98 |
|
Asia excl. CN, JP, KR |
1,083,641 |
2,700,407 |
743 096 |
0.40 |
1.46 |
|
Japan |
423,684 |
128,499 |
20 095 |
3.30 |
21.08 |
|
North America |
289,271 |
346,251 |
72 501 |
0.84 |
3.99 |
|
Europe |
173,862 |
737,851 |
144 163 |
0.24 |
1.21 |
|
South America |
119,079 |
597,995 |
158 510 |
0.20 |
0.75 |
|
Oceania |
30,820 |
37,498 |
8 625 |
0.82 |
3.57 |
|
Africa |
9,631 |
1,066,410 |
295 584 |
0.01 |
0.03 |
- In the original manuscript, we wrote that “This would suggest that the cultural proximity argument cannot fully explain how K-pop came to enjoy significant popularity at a global level.” In other words, we did not completely deny cultural proximity, rather we argued that there are more logical factors that explain better the global emergence of K-pop. Furthermore, knowing that K-pop was initially promoted in Asian countries, the question would then be on whether its popularity in the region is due to marketing activities or cultural proximity. But this question is beyond the scope of this paper.
- Regarding Table 1, Reviewer 1 argues that this table proves the cultural proximity argument as it shows that the popularity of K-pop is related to the diaspora communities or the number of Korean diaspora around the world. However, due to great contrasts among population sizes and demographics around the world that on its own does not accurately reflect the real popularity of K-pop. In this respect, Columns [4] and [5] are more important and meaningful than Column [1].
- If we only consider the cultural proximity aspect, the culture of North America and Oceania are far from that of Korea. Asia is closer, specifically Northeast Asian countries. The Asian data in Table 1 excludes data of China.
- In order to avoid possible confusion, we have revised the text as follows.
Before:
Jung and Song (2012) present the number of views on YouTube by region based on data from 2011, a year before the viral spread of “Gangnam Style.” The fact that a large portion of views are concentrated in Asia would seem initially to support the cultural proximity argument (see Table 1). Although the absolute number is meaningful, the analysis can be different if we consider the number of views among the young generation of each country as they are more likely to be the consumers of K-pop. Such an approach is used due to great contrasts among population sizes and demographics around the world that on its own does not accurately reflect the real popularity of K-pop. The new approach used here thus reveals a different picture. Compared to those in Asia, people in North America and Oceania are actually the consumers who viewed K-pop videos the most. This becomes even more evident when focusing on the age group of 15-24 who are considered to be the main consumers of K-pop. This would suggest that the cultural proximity argument cannot fully explain how K-pop came to enjoy significant popularity at a global level.
After:
Jung and Song (2012) present the number of views on YouTube by region based on data from 2011, a year before the viral spread of “Gangnam Style.” The fact that a large portion of views are concentrated in Asia would seem initially to support the cultural proximity argument (see Column [1] of Table 1). Although the absolute number is meaningful, the analysis can be different if we consider the number of views per person and also per youngster of each country as they are more likely to be the consumers of K-pop; in other words, data for Columns [4] and [5] are more important than that of Column [1] for a fair comparison. Such an approach is used due to great contrasts among population sizes and demographics around the world that on its own does not accurately reflect the real popularity of K-pop. This new approach used here thus reveals a different picture.
Table 1. A new interpretation on the number of YouTube views for K-pop (2011).
|
Regions |
No. of views (2011, Unit: 1,000)a [1] |
Populations (2011, Unit: 1,000)b |
No. of views/person |
||
|
Total [2] |
Age group (15-29 y.o.) [3] |
Total [4]=[1]/[2] |
Age group (15-29 y.o.) [5]=[1]/[3] |
||
|
Asia (excl. China and Korea) |
1,507,325 |
2,828,906 |
763,191 |
0.53 |
1.98 |
|
Asia excl. China, Japan, and Korea |
1,083,641 |
2,700,407 |
743,096 |
0.40 |
1.46 |
|
Japan |
423,684 |
128,499 |
20,095 |
3.30 |
21.08 |
|
N. America |
289,271 |
346,251 |
72,501 |
0.84 |
3.99 |
|
Europe |
173,862 |
737,851 |
144,163 |
0.24 |
1.21 |
|
S. America |
119,079 |
597,995 |
158,510 |
0.20 |
0.75 |
|
Oceania |
30,820 |
37,498 |
8,625 |
0.82 |
3.57 |
|
Africa |
9,631 |
1,066,410 |
295,584 |
0.01 |
0.03 |
Notes: 1. On the number of views, the original source does not clarify if the Caribbean region is included in the category of “South America.” Therefore, in contrast to the original format, the population of the Caribbean region is integrated into South America; 2. Data on China is not included in the Asia category due to unavailability of YouTube in the country; 3. The number of views in Korea and Korean population are excluded.
First, compared to those in Asia (excluding China, Korea, and Japan), the people in North America and Oceania are actually the consumers who viewed K-pop videos the most when the number of views per capita is taken into account (see Column [4]). It is noteworthy to mention that this data is from the period before the success of Psy and BTS and that Japan has been the main target for K-pop bands. Second, this becomes even more evident when focusing on the age group of 15-29 who are considered to be the main consumers of K-pop. When Japan is excluded, youths in North America and Oceania consume more K-pop than the same age group in Asia, again when the number of views per capita is considered (see Column [5]). In order to better understand this perspective, it is also necessary to point out the fact that Korean entertainment companies have promoted K-pop more in Asia, notably in Japan, during its early days. Furthermore, Psy and BTS have gained more popularity in North America than elsewhere. All of these explanations suggest that the cultural proximity argument cannot fully explain how K-pop came to enjoy significant popularity at the global level.
Some may argue that the data from Jung and Song (2012) presented in Table 1 does not consider the influence of the Korean diaspora. However, this point is insignificant when taking into account the ratio of diaspora which we have calculated additionally (see Table 2). In this respect, using data for 2011 from the Ministry of Foreign Affairs of the Republic of Korea (2016), the ratio of Korean diaspora in each region’s population appears to be almost 0 percent. Among these regions, North America has the highest diaspora population which is 0.666 percent of the total population while Oceania, Europe, Asia (excluding China, Japan, and Korea), South America, and Africa reach almost 0 percent; 0.429, 0.089, 0.043, 0.019, and 0.001 percent, respectively.
Table 2. Presence of Korean diaspora (2011).
|
Regions |
Total population (2011, Unit: 1,000)a |
Total no. of Korean diasporab |
Ratio of Korean diaspora |
|
Asia (excl. China and Korea) |
2,828,906 |
1,205,479 |
0.043% |
|
Asia excl. China, Japan, and Korea |
2,700,407 |
292,382 |
0.011% |
|
Japan |
128,499 |
913,097 |
0.711% |
|
N. America |
346,251 |
2,307,082 |
0.666% |
|
Europe |
737,851 |
656,707 |
0.089% |
|
S. America |
597,995 |
112,980 |
0.019% |
|
Oceania |
37,498 |
161,038 |
0.429% |
|
Africa |
1,066,410 |
11,072 |
0.001% |
Notes: 1. The regional distinction is adopted from Table 1; 2. Data for age groups of Korean diaspora is unavailable.
Data sources: a. Nations (2020); b. Ministry of Foreign Affairs of the Republic of Korea (2016).
Comment 6: traditional Korean pop music: It is already hybrid (Line 105, Page 3)
- The title of this paper is “The digital transformation of the Korean music industry and the global emergence of K-pop”. In this regard, we clearly stated that “candidate factors should be carefully analyzed by distinguishing changes that have happened ex ante or ex post for the international emergence of K-pop” and that “In this regard, this paper links digital technology to the music industry and focuses on identifying the key elements that have fostered the digital transformation of the Korean music industry, which has resulted in the global emergence of K-pop.” Furthermore, we specified that “There are various types of hybridity to consider here, but for this paper we will consider music style and lyrics.” Hence, the focus is on how different the situation was before and after the emergence of K-pop. Whether Korean traditional pop music was hybrid or not is beyond the scope of the paper.
- But, is there any culture that has not been hybridized before? If cultures of each country are all hybridized, then why are some of them more popular while others are not?
Comment 7: Existence of numerous hybrid musical practices does not mean that particular hybridity is not very important for certain genre (Lines 120-122, Page 4)
- In our paper, the section specifies the focus of “hybridity”; thus, music style and lyrics. And when compared with the music of other countries, hybridity cannot explain fully the emergence of K-pop. We also hinted that there are two steps: first, to reach the same competitive level with others; second, to outperform others. As we described in our literature review, some argue that hybridity was the key factor that made K-pop internationally popular. But their argument fails to answer the question on why K-pop has demonstrated better results than others? This becomes even more stark given the fact that hybridity is not a unique characteristic to K-pop. Evidently, the comment from Reviewer 1 seems to be more focused on the first step only.
- In order to avoid this confusion, we have modified the sentence as follows:
Before:
Even if it led K-pop to reach a competitive level with other prevailing music in the global market, it is still hard to argue that this hybridization per se makes K-pop internationally more popular than others.
After:
Even if it led K-pop to reach a competitive level with other prevailing music in the global market, it is still hard to argue that this hybridization per se makes K-pop internationally more popular than others that are also hybridized.
Comment 8: Brief description of K-pop fandom is needed here (Line 133, Page 4) and many fan groups (Line 146, Page 4)
- Thank you for this comment. We have revised the relevant section as follows:
Before:
Fandom. Several studies have identified the zealous fandom for K-pop as a factor in its international popularity (Jin and Ryoo, 2014; Min, Jin, and Han, 2018; Yoon, 2018). The importance of K-pop fandom and its activities to support their favorite bands and create a sense of community should not be overlooked, although a focus on fandom perhaps begs the question: how did they come to know about K-pop, how were they able to access the music? Above all, why were they interested in K-pop in the first place?
After:
Fandom. Several studies have identified the zealous fandom for K-pop as a factor in its international popularity (Jin and Ryoo, 2014; Min, Jin, and Han, 2018; Yoon, 2018). Fans of K-pop often open websites for various online activities to support their favorite bands. They routinely play newly released songs of their supporting band on various music sites. Some even externalize by posting billboards in public spaces to express support for their favorite bands or organize crowd funding projects. The importance of K-pop fandom and its activities to support their favorite bands and create a sense of community should not be overlooked, although a focus on fandom perhaps begs the question: how did they come to know about K-pop, how were they able to access the music? Above all, why were they interested in K-pop in the first place? In this respect, why are the fandom activities for K-pop significantly more visible than those for bands from other countries?
Comment 9: Porter (Line 161, Page 5)
- The full name already appeared before on page 2 in the revised version.
Comment 10: There is also participatory aspect in dance. (Line 364, Page 9)
- Thank you for the comment. We understand why you mentioned this point. However, this section focuses on the production side only as its title states “3.2.2. Producers: From audio sound to visual images”.
Comment 11: Table 2 (Page 12): explanations for “enhanced musicality”, “M&As” and “Chaebolization”
- Thank you for the comment. In fact, Table 2 is more like a summary of the existing analysis. Musicality is already explained at Line 282 on Page 7.
- “M&A” is explained at Line 325 on page 8.
- “Chaebolization” is at Line 427 on Page 11. But for clearer understanding we have modified as follows:
Before:
Such a diversification in business operations has pushed these entertainment companies to target stars who have differentiated talents that fit these varying sectors more effectively. To meet the new market demands, these companies have placed more effort and investment into fostering promising groups and singers by utilizing revenues gained from previously successful acts within the same company (Hong, 2012; Oh and Park, 2012). For example, with the profits earned by successful singers and groups such as BoA and Girls’ Generation, SME was able to invest in future talent. In the same way, entertainment companies have invested in diversified business by using revenues from various cash cow sectors. As time goes by, these businesses have shown a patchy performance and have had to be restructured in order to possess more efficient and synergetic networks.
After:
Such a diversification in business operations has pushed these entertainment companies to target stars who have differentiated talents that fit these varying sectors more efficiently. To meet the new market demands, these companies have placed more effort and investment into fostering promising groups and singers by utilizing revenues gained from previously successful acts within the same company (Hong, 2012; Oh and Park, 2012). For example, with the profits earned by successful singers and groups such as BoA and Girls’ Generation, SME was able to invest in future talent. In the same way, entertainment companies have invested in diversified business by using revenues from various cash cow sectors. This is similar to the chaebols or conglomerates that emerged in Korea as a way to interact with both related and seemingly unrelated sectors; thus “chaebolization.” As time goes by, these businesses have shown a patchy performance and have had to be restructured in order to possess more efficient and synergetic networks.
Comment 13: Consider: Hyunjoon Shin, Seungh-Ah Lee (eds.), Made in Korea: Studies in Popular Music, Routledge, 2017 (especially: Sun Jung, “Emerging Social Distribution: The Case of K-Pop Circulation in the Global Pop Market”).
- We added the reference “Jung. S. (2017), “Emerging social distribution: The case of K-pop circulation in the global pop market”, in Shin, H. and Jung, S. (Eds.), Made in Korea: Studies in Popular Music, Routledge, New York, pp. 61-72.
More importantly, we went over the whole article several times to make it more solid. Thank you for your great contribution.
Reviewer 2 Report
The paper concerns digital transformation of Korean music industry and the emergence of K-pop. The title and abstract are appropriate for the content of the text. It is a well-structured and clear paper. I think the authors should provide more details concerning the methodological approach applied in this paper. Furthermore, their analysis is well performed.
Author Response
We appreciate the constructive comments and encouragement from reviewers. They were very useful to help us improve the quality of this paper. We have incorporated most of the suggestions and hope that our responses and revisions will be satisfactory. What follows presents our responses and the changes we have made.
Comments and responses
Reviewer 2: The paper concerns digital transformation of Korean music industry and the emergence of K-pop. The title and abstract are appropriate for the content of the text. It is a well-structured and clear paper. I think the authors should provide more details concerning the methodological approach applied in this paper. Furthermore, their analysis is well performed.
- Thank you for the comments. We found that there are some confusing parts in the methodology section. Therefore, we have modified to make it clearer.
Before:
It can further explain how competitive advantages can be created, enhanced, and sustained. In particular, it is critical to pay attention to the strategies that create competitive advantages in a given environment even transforming disadvantages into advantages. In his study, Porter argues that companies or industries achieve competitive advantage through acts of innovation, either in new technologies and new ways of doing things, or in pursuing existing ideas that have never been vigorously pursued (p.45). As a result, companies have strategies to fit both industry dynamics and changing environments.
After:
This can further explain how competitive advantages can be created, enhanced, and sustained. In particular, it is crucial to pay attention to the point that competitive advantages can be created in a given environment and disadvantages can even be transformed into advantages (Moon, 2016; Porter, 1990). In his study, Porter argues that companies or industries achieve competitive advantage through acts of innovation, either in new technologies and new ways of doing things, or in pursuing existing ideas that have never been vigorously pursued (p.45). As a result, strategies (or industry and business activities) can develop to fit both industry dynamics and changing environments.
- Regarding the conclusion, it may seem to be not well supported but this is due to the arrangement of this paper. We put the results of the analysis in the discussion section in order to formulate a framework. The two sections, discussion and conclusion, are designed to supplement each other.
More importantly, we went over the whole article several times to make it more solid. Thank you for your great contribution.
